# Sustainability of coral reefs are affected by ecological light pollution in the Gulf of Aqaba/Eilat

Yael Rosenberg [1], Tirza Doniger [1] & Oren Levy[1]

As human populations grow and lighting technologies improve, artificial light gradually alters natural cycles of light and dark that have been consistent over long periods of geological and evolutionary time. While considerable ecological implications of artificial light have been identified in both terrestrial and aquatic habitats, knowledge about the physiological and molecular effects of light pollution is vague. To determine if ecological light pollution (ELP) impacts coral biological processes, we characterized the transcriptome of the coral *Acropora eurystoma* under two different light regimes: control conditions and treatment with light at night. Here we show that corals exposed to ELP have approximately 25 times more differentially expressed genes that regulate cell cycle, cell proliferation, cell growth, protein synthesis and display changes in photo physiology. The finding of this work confirms that ELP acts as a chronic disturbance that may impact the future of coral reefs.

---

[1] Mina and Everard Goodman Faculty of Life Sciences, Bar-Ilan University, Ramat Gan 52900, Israel. Correspondence and requests for materials should be addressed to Y.R. (email: yaelirose@gmail.com) or to O.L. (email: oren.levy@biu.ac.il)

Life on earth has evolved under constant environmental change[1,2]. The daily, tidal, lunar, and annual cycles are the most pronounced, affecting different life forms. Most organisms have developed an endogenous clock that allows them to anticipate daily and seasonal changes and adapt their physiological, behavioral, and biochemical activity accordingly[3,4]. Endogenous clocks are entrained to their local conditions by environmental cycles through input cues such as light or nutrition. Rhythmic events are common in biological systems and range in frequency from seconds to many years[1]. Environmental and behavioral conditions, such as physical activities, stress, and timing of food intake, are known to disrupt circadian rhythms; however, the most potent factor disrupting the normal circadian rhythms is unsuitably timed light, particularly light at night (LAN)[5]. It is agreed that most organisms anticipate environmental changes using the solar and lunar cycle and the disruption of these rhythms due to artificial lighting has major implications on timing of reproduction, migration, feeding, foraging, sleeping, and activity that can all lead lower chances of survival[6].

Human populations on earth are expanding rapidly, and artificial light is central to the functioning of modern society. The introduction of artificial lighting, particularly electric lighting, has disrupted natural cycles of light and darkness and is now referred to as light pollution or ecological light pollution (ELP), a term that has been coined to describe all types of artificial light that alter the natural patterns of light and dark in ecosystems[7].

Rapid global increase in artificial light at night (ALAN) has been proposed to be a new threat to terrestrial and marine ecosystems. Recently, it was found that the vast majority of the pelagic community exhibits a strong light-escape response in the presence of artificial light, observed down to 100 m[8]. ALAN is rapidly spreading globally at an estimated rate of 6% per year[9]. It has been shown to affect the physiology and behavior of various organisms[10], with consequences for species, communities, and population dynamics[9,11].

Globally, coastal human populations are growing rapidly, leading to increasing light pollution[12,13]. The alteration of natural lighting regimes could be expected to have a crucial effect on marine organisms because light, along with temperature, structures aquatic habitats[14–16]. Despite the well-known and strong influence of light on the behavior of aquatic organisms, little research has addressed the consequences of human disruption of diel, lunar, and seasonal cycles of illumination[16].

Symbiotic corals are highly photosensitive and are likely to be susceptible to ELP as they are often found in shallow, clear water with relatively high light levels. In addition to the role of light in photosynthesis and calcification, specific types of light can impact numerous aspects of coral biology as many species synchronize their spawning through detection of low-light intensity from moonlight[14,17,18]. Corals have developed strong circadian and diel behaviors, including extension and contraction of polyp tentacles, extending at night to feed on plankton and retracting during the day when they receive carbon from their symbiotic algae[14,19]. Absorption of visible light by seawater is greatest for long wavelengths like red, and shorter for wavelengths including blue which penetrate deeper in the water column. Blue light plays a key role in coral growth, zooxanthellae density, Chlorophyll a content and photosynthesis rates[20,21]. Light detection in corals is mediated through light-sensing molecules such as cryptochromes (CRY)–proteins that convert light and cause changes in the intracellular levels of second messengers, typically calcium[22]. Since the life cycle of coral reefs is governed by light, there is no doubt that artificial light disturbs their natural processes. Nighttime lighting has no natural analog—sunlight, moonlight, and starlight are the only sources of light corals have adapted to[17].

Many impacts of artificial night lighting have been established and the extent of its effects is becoming evident[6]. The majority of studies dealing with the ecological effect of light pollution on the environment have focused on behavioral aspects and timing of biological activities and are well understood for some groups of organisms, mostly among mammals[6,7]. The fact that there is almost no organism immune to this form of pollution has led to a growing interest and focus.

The diversity, frequency, and scale of human impacts on coral reefs are increasing to the degree that reefs are threatened worldwide[23–26]. A major concern is that the rate of environmental change exceeds the evolutionary ability of coral species to adapt. Human caused ELP could alter the natural light regimes of coral reefs by causing persistent disturbance or even chronic stress. In this study, we conducted a 5-month experiment with natural and light pollution treatment on the coral *Acropora eurystoma*, using transcriptome and physiology analysis to determine the cellular impact of ELP on a common reef coral. We aimed to reveal specific cellular pathways and genes that cluster into functional groups which change between the treatments in response to the daily and moon phase cycles. Our results show the effect of ELP on corals gene expression, causing elevated differentially expressed genes in corals under light treatment that cluster into pathways regulating cell cycle and protein synthesis. Finding the variations in gene expression caused by the different light regime could help better understand the effect of light at night on corals life cycle.

## Results

**Physiological parameter comparison.** At present, our understanding of the molecular and cellular reaction of reef-building corals to nighttime lighting is far from complete. In this study, we used accurate light measurements from the reef (Table 1) and were able to establish an aquarium system that could mimic the duration and intensity of the light reaching the shallow reef in the Gulf of Eilat where corals were collected (4–5-m depth)[16]. Physiology assay ($n = 5$ coral per treatment) showed no significant difference in symbiont cell concentration ($t$-test $_{(8)} = -1.187$, $P$ value = 0.2676) normalized to surface area nor in total protein concentration (mg total protein cm$^2$ ($t$-test $_{(9)} = 1.330$, $P$ value = 0.2175) between the two experimental groups (Fig. 1a, b, respectively). All Chlorophyll parameters were significantly different between the groups and had higher concentrations in the ELP samples, total Chl-a concentration normalized to symbiont cell (pg total Chl cell$^{-1}$), total Chl-c2 concentration normalized to symbiont cell (pg total Chl cell$^{-1}$) and total Chl-a concentration normalized to surface area (μg total Chl-a cm$^2$) (Fig. 1c–e).

**Differential expression analysis.** Next we used analyses of RNA-Seq and generated a de novo transcriptome for the coral *Acropora eurystoma* to evaluate changes in gene expression caused by the different light regimes. Gene expression patterns revealed an increase in differentially expressed genes ($P$-value < 0.05 and fold change > 1.5) in samples taken from ELP corals when compared with samples taken from ambient light (AMB)-exposed corals (8050 DE genes and 315 DE genes, respectively). To evaluate, which experimental variable (light and moon condition) had a greater effect on expression patterns of differentially expressed genes, we arranged all samples by groups according to the different conditions. Moon phase appeared to have less impact, in contrast to the light treatment which caused a strong and clear pattern of gene expression in each experimental group (Fig. 2a), with the ELP samples again having a higher number of differentially expressed genes compared with the AMB samples.

When comparing the gene variability within the experimental groups we could see that the AMB samples showed higher variability across the different sampling times (Fig. 2b), and a

**Table 1 Measurements of light, temperature, and pH from one aquarium per treatment and the northern part of the Gulf of Aqaba/Eilat in the Red Sea**

**Full moon 13/6/2014**

| | Temperature Max (°C) | Temperature Min (°C) | Nighttime PAR ($\mu mol\ m^{-2}\ s^{-1}$) | Day time PAR ($\mu mol\ m^{-2}\ s^{-1}$) | pH |
|---|---|---|---|---|---|
| *Full moon 13/6/2014* | | | | | |
| Open water northern shore (sea surface) | 25.4 | 23.9 | 5 | 1971 | N/A |
| AMB | 25.5 | 24 | 0 | 1580 | 8.15 |
| ELP (water top) | 25.5 | 24 | 4.8 | 1580 | 8.15 |
| *New moon 25/6/2014* | | | | | |
| Open water northern shore (sea surface) | 25.6 | 24.2 | 5 | 1967 | N/A |
| AMB | 25.6 | 24.5 | 0 | 1573 | 8.2 |
| ELP (water top) | 25.6 | 24.5 | 4.8 | 1573 | 8.2 |

Upper part is data measured during the first sampling day (full moon), lower part is data measured during the second day of sampling (new moon). Light unites refer to the quanta of light in parabolic anodized reflector (PAR) range measured

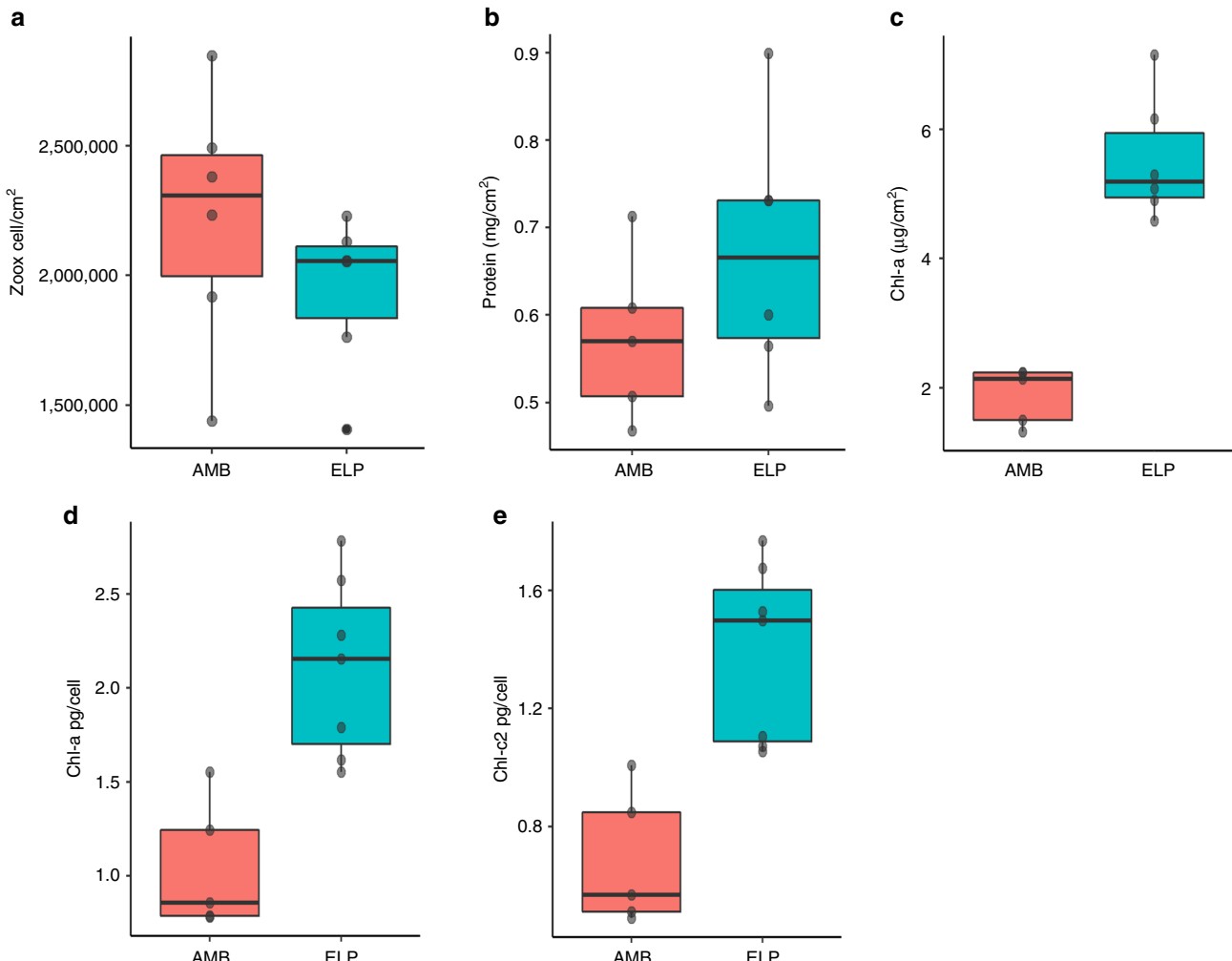

**Fig. 1** Physiological parameters repeatedly measured in all colonies on the last sampling day. **a** Symbiont cell density. **b** Total protein concentration. **c** Total symbiont Chlorophyll normalized to colony surface area. **d** total symbiont Chlorophyll a normalized to symbiont cell. **e** Total symbiont Chlorophyll-c2 normalized to symbiont cell. Red bars represent ambient (AMB) samples, light blue bars represent ecological light polluted (ELP) samples

slight difference between the two moon phases at the same time of sampling on the different sampling days. The ELP samples did not express a noticeable change regarding the moon phase. A variability increase was observed during noon and sunrise samples (Fig. 2c), although the variability was lower than the AMB samples. We further noticed that in both experimental conditions samples from midnight and sunset had low variance in gene expression and higher variance during the noon sample, i.e., strongest illumination throughout the day causes strong variance in gene expression between samples.

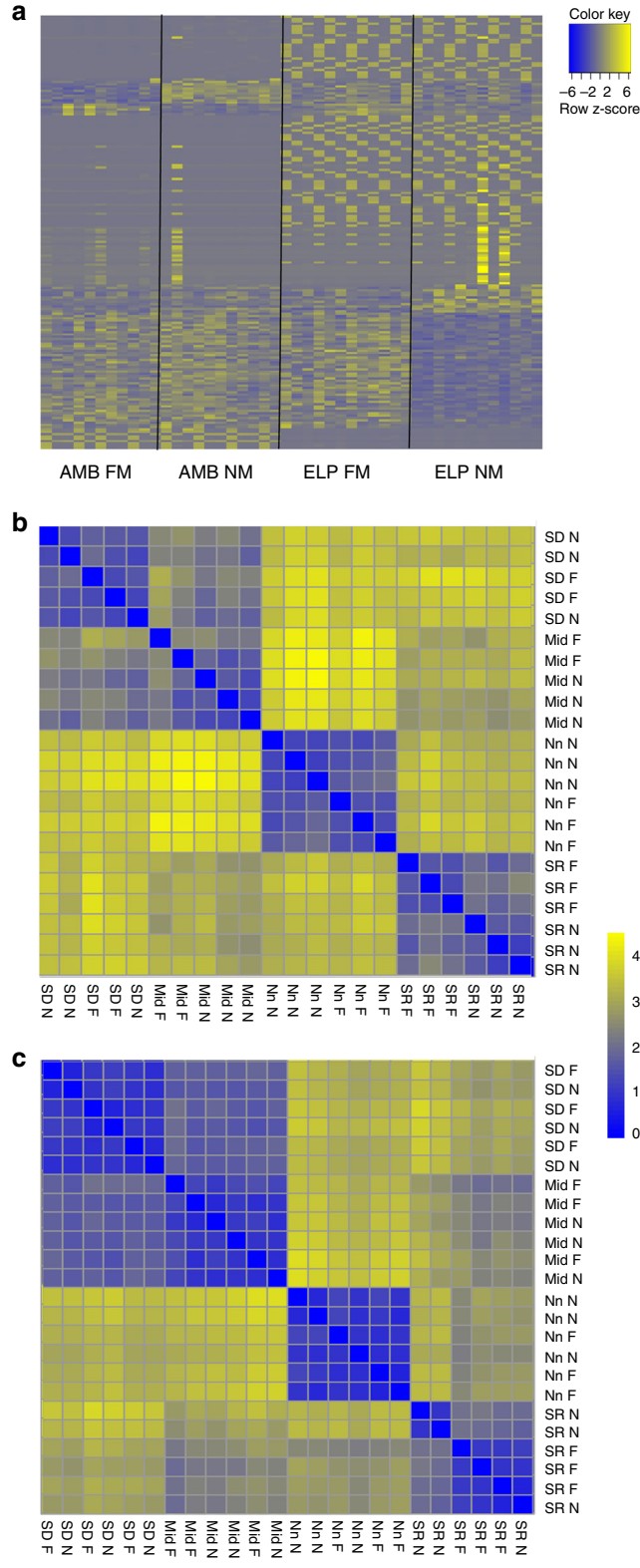

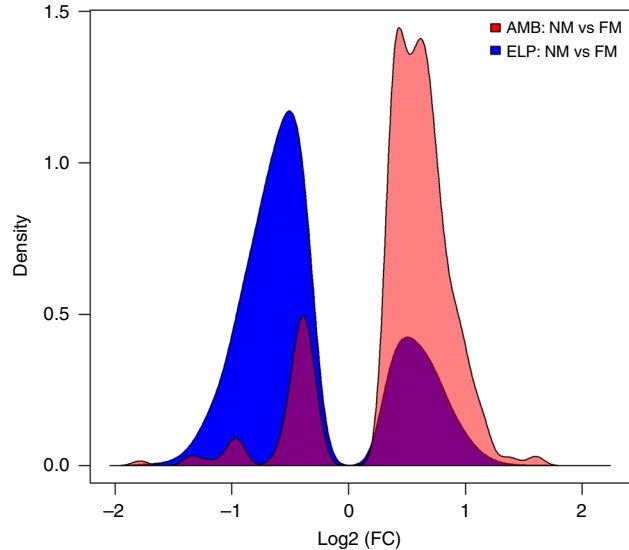

**Fig. 2** Differential expression of transcripts for the two different experimental conditions comparing reaction with light regime. **a** Heatmap of all differentially expressed transcripts with hierarchical gene clustering divided by experimental groups (AMB—control corals, ELP—light polluted treatment) and moon phase (new moon—NM, full moon—FM) in order to reveal expression patterns and effect of experimental variables (light regime and moon phase). Expression level is indicated by z-score, each line is an individual gene. **b** Heatmap presenting distance matrix of all AMB samples against each other showing the gene variability within the group. **c** Heatmap presenting distance matrix of all ELP samples against each other showing the gene variability within the group. SD: sun-down, SR: sunrise, Nn: noon, Mid: midnight, N: new moon, F: full moon

**Fig. 3** Kernel-density chart comparing between new moon (NM) and full moon (FM) in each experimental condition. Pink indicates AMB samples, blue indicates ELP samples, purple indicates those genes from one group whose expression pattern resembles the genes from the opposite group. Genes are divided by Log2 fold-change and the density is the probability that the variable will fall within the curve. Upregulated genes are aligned to the right, downregulated genes are aligned to the left

**Functional molecular pathway analysis**. By analyzing the differentially expressed genes from each condition, and comparing the genes expressed by all new-moon samples in comparison with full moon we found a specific regulation pattern for each experimental group. The AMB differentially expressed genes were clearly upregulated during new moon while the ELP differentially expressed genes were strongly downregulated during new moon (Fig. 3).

Canonical pathways that are linked to light pollution treatment were found by comparing the differentially expressed genes of the ELP samples vs. the AMB samples at each sampling time during the day. This comparison revealed many different pathways that are differentially expressed only in corals that were exposed to 4 months of ELP. Most of these pathways are associated directly to cell-cycle control, cell growth and cell division (pathways $P$-value 4.96E−07). Prior knowledge on these pathways associates them to cancer ($P$-value 2.58E−05, 2625 molecules) and reproductive system diseases ($P$-value 5.32E−06, 1654 molecules)[27–31]. Further investigation of the pathways and examination of gene networks within the pathways revealed that they all share some relevancy to a specific group of genes (Fig. 4). This group of genes includes PI3K (phosphoinositide 3-kinase), AKT (protein kinase B), and mTOR (mechanistic target of rapamycin) which create signaling pathways regulating cell cycle and cell survival[32–34].

## Discussion

To better understand the increasing effect of ALAN on marine areas such as coral reefs which reside in shallow waters, we conducted a 5-month comparative experiment on the coral

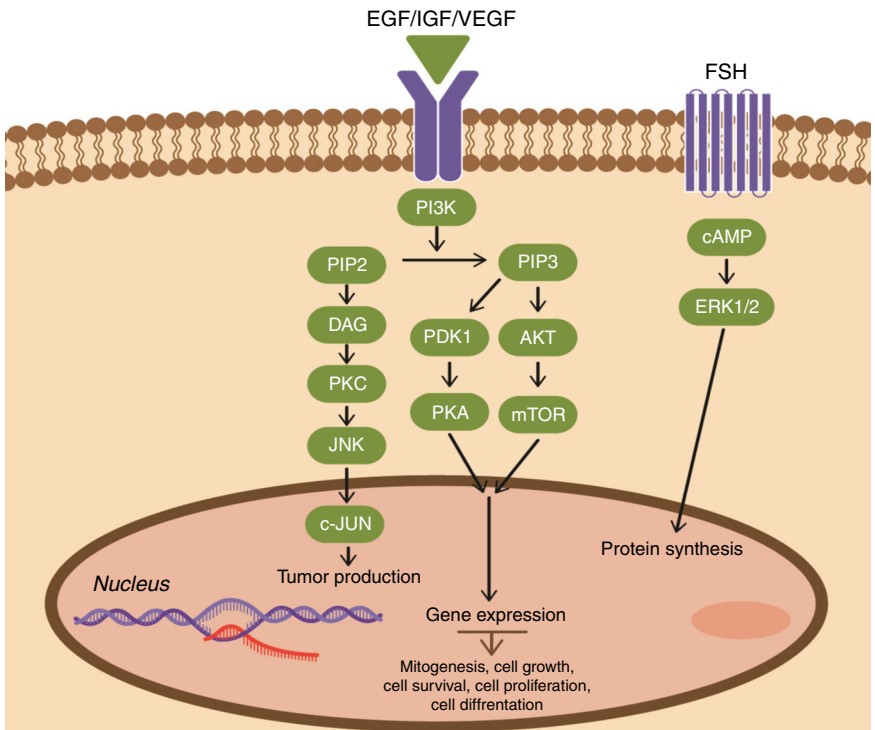

**Fig. 4** proposed pathway for coral colonies under ELP condition containing known genes involved in cell growth, proliferation, survival, and differentiation that were found in our study to be differentially expressed and upregulated in corals under ecological light pollution (ELP). PI3K: phosphoinositide 3-kinase, PIP2: phosphatidylinositol (4,5)-bisphosphate, PIP3: phosphatidylinositol (3,4,5)-trisphosphate, DAG: diacyl glycerol, PKC: protein kinase C, JNK: c-Jun N-terminal kinase, c-JUN: c-JUN gene, PDK1: pyruvate dehydrogenase kinase 1, AKT: protein kinase B, PKA: protein kinase A, mTOR: mammalian target of rapamycin, cAMP: cyclic adenosine monophosphate, ERK1/2: extracellular signal-regulated kinases 1/2, FSH: follicle-stimulating hormone, VEGF: vascular endothelial growth factor, IGF: insulin-like growth factor 1, EGF: epidermal growth factor

*Acropora eurystoma* under two light regimes. Corals under AMB conditions were used as control to study the effect of light at night on coral physiology and gene expression. Corals under ELP treatment were placed under light conditions matching the light pollution experienced by corals in the northern areas in the Gulf of Aquaba/Eilat. An emphasis was made to create an experimental setup that would resemble the same environmental conditions as the light contaminated areas in the natural reef. A comparative analysis between corals under natural light cycles and corals under ELP treatment led us to identify unique pathways that are activated or repressed under light pollution conditions that result from the possible disruption of the biological clock and can lead to abnormal cell proliferation and growth.

Regulation of diurnal and circadian rhythms and cell proliferation are linked in many organisms from cyanobacteria to humans[5,35]. Exposure to light at night causes circadian disruption that can alter the regulation of circadian genes and hormones and, in turn, alter physiology and metabolism, leading to increased long-term risk for the development and promotion of diseases[36,37].

Physiology results suggest that exposure to ELP creates an environment with alternating phases of natural light during the day and constant low-level artificial light during the night. These light conditions, as opposed to constant light with no change in spectrum and intensity, have hardly been studied. The physiological assay was conducted in order to estimate differences in both experimental groups. The results show variance between treatments regarding the photo-physiology parameters resulting in higher Chlorophyll concentration in ELP samples as opposed to AMB samples (Fig. 1c–e). The light/dark cycle regulates many cellular processes, such as chloroplast differentiation, DNA repair, cell division, embryogenesis, and gametogenesis[38], and a

dark period is crucial for stress recovery and repair[10]. Under continuous light, the known clock genes in corals were shown to express arrhythmic profiles[22,39–42] and this may be also occurring in algae[43]. The disruption of circadian clocks and dependent physiological and developmental processes might explain the observed elevated Chlorophyll concentration with no change in algae count under ELP conditions. The replacement of the natural dark phase with low-light illumination, ELP may provide conditions equivalent to those of continuous illumination, with two varying phases of light intensity—the sunlight at day and artificial light at night. Continuous light can have both positive and negative effects on plants and microalgae for reasons that are still not fully understood nor studied[44].

Generally, our findings show, molecular evidence in corals of the effects of exposure to light at night that match those of more complex organisms, mainly mammalians. Using transcriptome analysis under the ALAN treatment (Fig. 5) to compare corals growing under natural light cycles and under light pollution conditions, we found many pathways that are altered. All ten coral colonies tested had the same conditions in our open seawater system (water temperature, sunlight, pH) except the exposure to light at night of one group (Table 1 and supplementary figs. 2–3), suggesting that these alterations in gene expression are a direct result of the exposure to light pollution. The top disorders (highest number of genes involved) in the treatment group in comparison with the control group include cell proliferation (2625 genes), organismal injury and abnormalities (2662 genes), and reproductive system disease (1654 genes).

Our analysis revealed many altered pathways that result in cell-cycle progression, cell proliferation, survival and growth. One notable pathway is the Insulin-like growth factor 1 (IGF-1) signaling. The type 1 Insulin-like growth factor receptor (IGF-1R) is

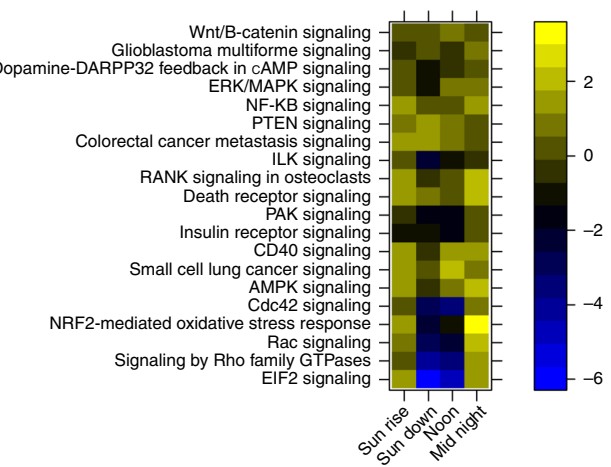

**Fig. 5** Gene ontology enrichment analysis for ELP genes from both moon phases, divided by sampling times, showing the different biological processes operating at each time point based on z-score. The color scale indicates the activity of the process

a known receptor in mitogenesis, transformation, and protection from apoptosis. IGF-1 causes an increase in the expression of Vascular endothelial growth factor (VEGF), a well-established central regulator of lymphangiogenesis. The way IGF-1 regulates the expression of VEGF through the P13K/Akt and mitogen-activated protein kinases/mitogen-activated protein kinases 1/2 (MAPK/ERK1/2) signaling pathways has been identified previously in our work regarding coral spawning[45,46]. In most of the altered pathways identified we found a specific group of genes which creates a signaling pathway regulating cell cycle and cell survival. Pathway initiation begins with the activation of PI3K through cell adhesion molecules, tyrosine kinase growth factor receptors, insulin-like growth factor receptors, G-protein coupled receptors, and oncogenes. The activated PI3K causes the formation of PIP2 and PIP3 that bind to the serine/threonine kinase AKT, which in turn activates mTOR and causes protein synthesis and cell growth[47,48]. This specific pathway was found in our coral samples under ELP treatment in many of the altered biological functions, as a core regulator in their signaling pathways (Fig. 4).

Many studies in vertebrates suggest that the PI3K signaling pathway is vital to the growth and survival of cancer cells[49–51]. Cancer research studies on corals have focused on tumor formation and composition only[52–54], though recently it has become evident that in pre-bilaterian animals there is natural occurrence of tumors, and it is predicted that most metazoans might be prone to develop tumors[55].

A number of recent studies have discussed light pollution as a potential risk factor for human breast and prostate cancers[56–59] due to the fact that light disrupts normal circadian timing. Since light/dark cycles are a basic component of the daily and annual temporal organization of any organism, it is not surprising that exposure to light pollution disrupts this temporal organization[1]. Genes that drive the circadian rhythm are emerging as central players in gene regulation, particularly for cell-cycle regulatory genes and the genes regulating apoptosis[60–62], and disruption of the normal biological rhythm is therefore associated with such pathological conditions. It is often assumed that cancer in animals developed during the evolution of the multicellular life style but it is now known that cancer is an ancient phenomenon with a direct correspondence to basic cell function and can be found in early branching metazoans[63].

Our results suggest that corals possess similar cellular reaction to light pollution as later evolved organisms, the altered light

regime causes cell proliferation and organismal abnormalities. The negative effects of ALAN has been demonstrated in many species across the tree of life[64]. Since coral reefs are typically found in shallow waters they are highly affected by the light/dark cycles. Many of their behaviors and biological processes are governed and synchronized by sun and/or moonlight[14,45,65,66]. Previous studies of light manipulation on corals have resulted in asynchronous planula release from different coral species[18,65]. The result of asynchronous reproduction, caused by ecological speciation, could lead to reproductive isolation and prevent gene flow between differentially lit coral communities[67]. The fact that such a small change in light exposure produces such a considerable effect on coral biology and timing suggests that light pollution could be a concern for shallow coral reefs globally.

Worldwide human population growth influences organisms through urbanization, industrialization, and transportation infrastructure[68]. The environmental disruption linked with the continuous increase in human populations has led to species extinction, altered community structure, and degraded ecosystem function[69]. Anthropogenic light pollution has become an increasingly important stressor, and the diversity of affected taxa grows and includes birds[70], rodents[13], ants[71], turtles[72], bats[73], plants[74] and many more.

Despite the globally widespread, intensifying and changing nature of light pollution, and its apparent severity, ALAN has not attracted the same levels of concern and interest from the scientific community and governments as other global change phenomena[75].

The ability of organisms to adapt rapidly to the introduction of light at night through behavioral, genetic or epigenetic changes is likely to be far more limited than for climate warming, due to the unprecedented nature of this change[69]. We have provided a perspective for the understanding of how coral reefs are threatened by this underappreciated driver of global change. Although our corals did not show any visible signs of bleaching, we postulate by our molecular analysis that EPL may act on coral life style as a chronic environmental disruption, whose impacts on growth and reproduction will manifest over a longer time scale (years). Therefore, we believe that corals under light pollution would not be able to perform their normal cyclic behaviors, eventually leading to global reef decline. We will continue with studies that will help determine the specific threshold of light intensity and quality (monochromatic) that effect corals, in hope to influence management decisions on optimal levels of light in coastal and urbanized areas. Future studies should be conducted over longer periods in order to better understand and characterize the EPL disturbance on coral reefs and marine environments.

## Methods

**Coral collection, maintenance, and sampling**. Corals were collected under the Israel Nature and National Parks Protection Authority permit number 2014/40227 issued to Y.R. and O.L. Ten mature colonies of *Acropora eurystoma* (measuring > 50 cm in diameter) were collected from 4–5-m depth from the coral nursery of the Inter-University Institute (IUI) at the Gulf of Aqaba/Eilat Red Sea (28.6929°N, 34.7299°E) in February 2014. This location was chosen since it has no light contamination during nighttime. Corals were divided in two groups and placed in flow-through aquarium systems in IUI in Eilat. The aquarium systems are part of the Red Sea Simulator (RSS), an outdoor system that is exposed to full spectrum sunlight (1800 µmol m$^{-2}$ s$^{-1}$ PAR at midday on a cloudless day) with a 15 h photo-period on the longest day and 11 h on the shortest day[76]. Seawater temperature and pH (Polilyte Plus H Arc 120, Hamilton) were monitored and tested every hour through automated monitoring (sensor-carrying robot) to ensure all corals were under the same conditions (Table 1 and Figs. 2–3). Each experimental group, consisting of five coral colonies, was under a specific light regime; the first was natural light cycles and moon phases (AMB corals). The second group had artificial light contamination (ELP corals) from small white LED light strips 6000–6500 K (4.8 mol quanta m$^{-2}$ s$^{-1}$ at the top of the aquarium and 1.5–2 µmol quanta m$^{-2}$ s$^{-1}$ at coral height) that were placed behind the glass walls and were turned on every day at sunset. Light intensity in the ELP treatment was carefully

adjusted to mimic the same levels of light during the nighttime that coral reefs in Eilat are exposed to based on Tamir et al.[16]. Light was measured using a LI-COR underwater quantum sensor LI-193, spectrum measurements were made using Ocean optics JAZ spectrometer (supplementary Fig. 1). In all aquarium systems sunlight was reduced with a neutral density filter (LEE Filters no. 210) based on Levy et al.[77], and was equal to the light intensity that penetrates the water at 5-m depth at the northern part of the Gulf of Aquaba/Eilat in the Red Sea[16]. A black plastic sheet was placed between the two aquarium systems to prevent light contamination between treatments. Corals were kept under experimental conditions for 4 months before sampling started at the full moon day in June 2014. Second sampling day was during the new moon day of June 2014. At each sampling day there were four sampling times: sunrise, noon, sunset, and high moon. In total there were two sampling days, 13 June 2014 (full moon) and 25 June 2014 (new moon) and eight sampling times, four for each sampling day.

Prior to each collection, a container was filled with liquid nitrogen and brought to the sampling area. A small branch from the top of each colony, including the axial polyp, measuring an average of 5 cm in length, was sampled from each coral using pliers. The sampled branch was placed in a small piece of aluminum foil with a tag containing the sample ID, time and colony, snap frozen in liquid nitrogen and transferred to a −80 freezer. During the entire experiment corals were evaluated daily and no signs of bleaching or excess algal cover were noticed.

**Physiology.** Coral fragments from the last sampling day ($n = 5$ for each group) were tested for protein concentration, zooxanthellae density, and Chlorophyll concentration to assess the health of the corals. Tissue was removed from frozen coral fragments using an airbrush and ice-cold filtered (0.22 uM) seawater. Skeletons were retained for surface area determination using the wax dip technique[78]. Tissue samples were homogenized for 30 s using an electrical tissue homogenizer (Kinematica Polytron™ PT2100 Benchtop Homogenizer). A sub-sample (100 μL) of the supernatant was taken to determine host protein concentration by a colorimetric method[79] using a multi-scan spectrum spectrophotometer (595 nm, 450 nm, Biotek HT Synergy plate reader) and bovine serum albumin as a standard (Quick Start Bradford Protein Assay, BIO-RAD). Protein concentration was used as a biomass and normalization index for the coral fragments. Samples were centrifuged to separate host and symbiont tissues. Further centrifugation and washing with filtered seawater was performed to isolate symbiont cells for cell counts (hemocytometer) and photosynthetic pigment extraction. Pigments were extracted for 24 h in 90% acetone at 4 °C in the dark and Chlorophyll (Chl) a and c2 concentrations were measured spectrometrically at 630, 664, and 750 nm with a multiskan spectrum microplate spectrophotometer (Thermo Fisher Scientific, USA). Chlorophyll concentration was determined as previously described[80] and normalized to zooxanthellae cells and surface area. Zooxanthellae were counted with a hemocytometer under a microscope and normalized to coral surface area. Each result is an average of five fragments per treatment. Physiology assay results of both treatments were compared with each other in each parameter using the R package unpaired two-samples t-test to represent statistical relevance.

**RNA extraction.** Total RNA was extracted from all 80 fragments using TRIzol reagent (Invitrogen) and a modified version of the manufacturer's protocol that included an additional chloroform extraction and magnesium chloride precipitation overnight. A small branch was cut off and placed in pre-cooled aluminum foil. The branch was crushed into a fine powder with a hammer, while the foil packet was occasionally dipped in liquid nitrogen to keep it frozen. The aluminum foil content was transferred to a 15 ml flacon pre-filled with 2 μL of TRIzol and left at room temperature for 5 min. The tubes were then centrifuged at $7500 \times g$ for 5 min at 4 °C to remove the skeleton powder. In total, 1500 μL of the supernatant from each sample was transferred to a 2 mL tube with 300 μL of chloroform, shaken vigorously and kept at room temperature for 10 min, followed by centrifuging at $12,000 \times g$ for 15 min at 4 °C. After this centrifugation, there were two visible phases. Eight hundred microliters of the aquatic phase was transferred to a 2 mL tube containing 600 μL of chilled isopropanol. Following a 10-min incubation at room temperature, the tubes were centrifuged at $12,000 \times g$ for 10 min at 4 °C. The remaining supernatant was removed and the visible pellet was washed with 1 mL of 75% ethanol and then centrifuged at $7500 \times g$ for 5 min at 4 °C. The final step of the ethanol wash was performed a second time and was followed by removal of the ethanol and drying of the tubes in a clean chemical hood. The dry pellets were covered in 500 μL RNAse free water and incubated at 57 °C for about 5 min until the pellet dissolved. Five hundred microliters of 5 M lithium chloride was added to the tube, gently mixed and stored in a −20 freezer overnight. The following morning, samples were defrosted on ice and centrifuged at $15,000 \times g$ for 30 min at 4 °C. The supernatant was removed without touching the pellet and 1 mL of 75% ethanol was added. Samples were centrifuged at $7500 \times g$ for 5 min at 4 °C and the supernatant was removed. The final step was again repeated, for a total of three ethanol washes. After the last wash, the ethanol was removed and the tubes were dried in a clean chemical hood. The dry pellets were covered in 40 μl RNase free water and incubated at 57 °C for about 5 min until the pellet dissolved. Purified RNA samples were analyzed using a NanoDrop 1000 spectrophotometer (ThermoScientific) to assess RNA quantity and a 2100 Bioanalyzer (Agilent) to assess RNA quality (RIN > 8.5).

**Next-generation sequencing.** From each sampling point we had three replicates sent for sequencing from each treatment. In total, 1.5-μg RNA from each of the 48 samples was sent for sequencing. RNA samples were prepared using the Illumina TruSeq RNA Library Preparation Kit v2, according to manufacturer's protocol. Libraries from each sampling point were run on lanes of an Illumina HiSeq2000 machine using the multiplexing strategy of the TruSeq protocol. Protocol starts with poly A selection that results in RNA selection only. Paired-end reads, 100 bases long, were obtained for each sample. All sequencing libraries were trimmed using TrimeGalore version 0.4.0 (http://www.bioinformatics.babraham.ac.uk/projects/trim_galore/) to remove adapters, primers, and low quality bases. All the libraries were merged and de novo assembly was performed using Trinity (version 2.2.0) with default parameters yielding 553,905 contigs (transcripts) with an N50 of 1406 and a median contig length of 415. The run_RSEM_align_n_estimate.pl script included with the Trinity software was used to map the reads for each library back to the assembled transcriptome and calculate sample-specific abundance for each transcript. In order to reduce the number of erroneous isoforms, all transcripts with an isoform percentage value (IsoPct%) < 1 were excluded from further analysis, yielding 545,469 contigs. These contigs were then further filtered for rRNA and other possible contaminations. BLASTP searches against the SILVA database yielded 416 rRNA matches. BLASTN searches against the NT database (100% identity, 90% query coverage) returned 174 bacteria, 76 *Symbiodinium*, and other non-*Acropora* matches. 553,114 contigs remained after filtering the assembly.

Putative coding regions were extracted from the transcriptome assemblies using TransDecoder software (www.transdecoder.sourceforge.net), with minimum length of 50 amino acids, providing all the CoDing Sequences and proteins from the assembly. The library contained 278,796 open reading frame (ORF) encoding contigs. Bowtie (v2.1.0) was used to map reads from each sample against these coding sequences. We performed a *Symbiodinium* specific search against a custom *Symbiodinium* database (symbs.reefgenomics.org) that revealed 21,992 transcripts that match *Symbiodinium* transcriptomes (80% query coverage and 90% identity and an e-value of $1^{e−10}$). In total, 66,589 coral transcripts were a match to the *A. digitifera* transcriptome. The remaining assembled contigs from coral and dinoflagellates were separated with Psytrans (https://github.com/sylvainforet/psytrans) using the *Acropora digitifera* v0.9[81] and *Symbiodinium goreaui* (Clade C, type C1)[82] (downloaded from http://symbs.reefgenomics.org/download/) predicted coding sequences as references.

Psytrans classified 101,801 transcripts as coral and 88,418 Symb transcripts. We compared the GC-content of the adi-matched, symb-matched, coral-classified, and symb-classified contigs using a kernel-density plot. The GC-content of the coral-classified and symb-classified transcripts matched their corresponding transcriptome (supplementary fig. 4). In total, we identified 168,390 A. eurystoma transcripts. Transcriptome completeness was assessed by comparing the contigs to the Benchmarking Universal Single-Copy Orthologue v. 2 (BUSCO)[83] with the metazoan orthologue set using the gVolante server[84]. The A. eurystoma transcriptome was found to be comprehensive, accounting for 94% of the core genes, and 97% if we include partial coverage. Note that the average number of orthologs per core genes is 1.8, meaning there are several transcripts that may represent one gene.

**Transcriptome annotation.** All sequencing libraries were trimmed and merged, and a de novo transcriptome was assembled for the *Acropora eurystoma* coral. Annotations were assigned by blasting the newly identified transcriptome against *Acropora digitifera*, *Homo sapiens*, Swissprot and UniProt50 database using BLASTP (NCBI). We filtered the BLASTP results in order to increase the certainty of obtaining true homologs. Annotation was assigned using the best matches obtained with an e-value threshold of $5 \times 10^{−5}$, $> = 30\%$ alignment identity, and $> = 70\%$ query coverage. In total, we were able to assign annotation to 119,625 (42%) contigs.

**Differential expression.** RNA-Seq results were analyzed using the R package GOSEQ (v1.18.0) to detect statistically significantly over-represented genes that cluster into functional groups, requiring a Benjamini–Hochberg-corrected P-value of ≤0.05 when comparing between samples from both experimental groups. Using the normalized data from the DESeq comparisons, we performed hierarchical clustering and generated heat maps of all the significantly expressed genes using the heatmap.2 function from the R BIOCONDUCTOR package GPLOTS (v2.17.0). Heatmap.2 was used with the default clustering method and scaling the data by rows. Genes with an adjusted P-value (Benjamini–Hochberg) of ≤0.05 and a minimum 1.5-fold change were considered differentially expressed. We began by comparing the expression profile of genes within the different treatments (ELP and AMB), to find gene groups that are differentially expressed during each moon condition and time of day, regardless of light condition. We continued by comparing the two experimental groups looking for different genes, for example, comparing between ELP and AMB samples during each moon condition, regardless of the time of day, to find differentially expressed genes responding to the moon phase; comparing between ELP and AMB samples during the day time sampling periods, regardless the moon phase, to find differentially expressed genes responding to the duration of the day, comparing all ELP samples to all AMB

samples in order to find specific pathways that are different between the two light regimes.

**Pathway analysis.** Pathway analysis of annotated sequences was performed using the IPA software (http://www.ingenuity.com). Only differentially expressed genes resulting from the DESEQ analysis were used as input for the software. The dynamic canonical pathways contained in IPA are well characterized metabolic and cell-signaling pathways that are compiled from the literature and the Kyoto encyclopedia of genes and genomes (KEGG). The IPA canonical pathways display the genes/proteins involved, their interactions, and the cellular and metabolic reactions in which the pathway is involved. Expression values were $z$-score normalized and a comparison was conducted in order to identify enriched pathways with a significant differential expression between the two experimental groups. All IPA results were filtered with a $P < 0.05$.

**Reporting summary.** Further information on research design is available in the Nature Research Reporting Summary linked to this article.

## Data availability

The sequencing data reported in this study has been deposited to the Sequence Read Archive (SRA), under accession SRP135621. All other data can be obtained from the corresponding authors.

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

## Acknowledgements

The research leading to this paper has received funding from the Israeli Science Foundation (ISF), grant number 3928 to O.L. We would like to thank the staff and students at the Inter-University Institute (IUI) for Marine Sciences in Eilat for their hospitality and assistance with the field work, and Prof. Maoz Fine for allowing us to conduct our experiment in the Red Sea simulator system. Light measurements and seawater temperature (SST) were assigned using the Israel National Monitoring Program at the Gulf of Eilat (http://www.iui-eilat.ac.il/Research/NMPMeteoData.aspx). This study represents partial fulfillment of the requirements for a PhD thesis for Y. Rosenberg at Faculty of Life Sciences Bar-Ilan University, Israel.

## Author contributions

Y.R. and O.L. designed the research and carried out the experiment. T.D. and Y.R. preformed data analysis. Y.R. and O.L. wrote first draft of the paper. All authors edited the paper. All authors read and approved the final paper.

## Additional information

**Competing interests:** The authors declare no competing interests.

