## [Peer Review File · Communications Biology]

Reviewers' comments:

Reviewer #1 (Remarks to the Author):

With the rapidly rising urbanisation of coastlines, marine ecosystems are exposed to accordingly increasing levels of artificial light at night (ALAN). As many important physiological processes in marine organisms are regulated and synchronised by solar and lunar cycles, there is a strong concern that Ecological Light Pollution (ELP) may interfere with those mechanisms. This holds true in particular for zooxanthellate reef corals, the habitat-forming species of warm water coral reefs. At present, the effects of ELP on corals are understudied and not well understood. Therefore, the presented manuscript provides important and timely insights in the interference of ELF with gene regulation patterns. Consequently, I am in favour of publication of this manuscript. However, its presentation leaves room for some improvement and I recommend a thorough revision of the paper.

Both, in the introduction and discussion, the logic flow can be improved. At present, the story switches several times between the "bigger/global picture" and specific aspects. Suggest to start with the bigger picture and then zoom in...

There is a strong focus of the manuscript, specifically in the discussion, on differentially regulated genes that are involved in cancer formation. I assume that many differentially regulated genes were identified to be homologous to genes involved in cancer formation in humans /model organisms because of the fact that many signalling pathways are best studied / named in the context of cancer formation. We don't know what the majority of these genes do in corals. While there are some reports on tumors in corals, I would recommend to steer clear from the conclusion that ELP may affect coral via disturbing cell proliferation / cancerogenesis. Instead, a much stronger emphasis should be placed on differentially regulated genes that have been previously studied in corals eg. in the context of spawning, circadian rhythms etc. What is going on with the CRYs?

In the introduction and discussion, the use of the experimental ELP intensity should be clearly explained and linked to the environmental data.

Figure display could be improved by increasing size/ legibility of axes labels and legend, removing "headers". In Fig. 4, there could be a better use of space (label / letter size) to improve legibility. Figure 5: Suggest to order "Sun rise – Noon – Sun down – Midnight"

The language and spelling of the text should be carefully edited.

The data and the annotation information should be deposited in suitable data base.

Specific comments:

Line 31: suggest "~25 times" instead of "95%"

Line 36: The focus should be taken away from the cancerous processes (See comment above).

Line 45: Include refs.

L52: re-phrase "dysregulation"

L55: Give examples for the "major implications"

L78: Include reference to blue light regulation of photoprotective pigments in coral host: D'Angelo C, Denzel A, Vogt A, Matz MV, Oswald F, Salih A, Nienhaus GU, Wiedenmann J. 2008 Blue light regulation of host pigment in reef-building corals. *Mar. Ecol. Progress Ser.* 364, 97–106.

L79: (CRYs) ???

L81: "structuring" does not work in this sentence, also the second half of the sentence is confusing – rephrase.

L84-85: Once define, suggest to stick with abbreviations "ALAN" and "ELP"

L90-94: Unclear –rework.

110: Ambient light cycle (AMB). Suggest to call those "control" vs "ELP". Stay consistent, throughout the ms there are various variations.

Justify the level of ELP used in the experiments clearly in the introduction.

L131: Give detail of the branch fragments. Where those the top 5cm incl. axial polyp? Further down?

L139: "electrically" – How?

L154: The RNA extraction is quite detailed and a full protocol is provided in the supplement. This should be shortened, highlighting the parts where the procedure deviated from the manufacturer's instructions.

L203 onwards: Check the genera / species names are set in italics.

L234: "proteome" ???

L288: "de-novo genome" ???

L.305: "express higher gene resemb." – rephrase

L.316: see comments on cancer

L329: "ecological marine areas" –rephrase.

L334: see comments on cancer

L336: Give examples of the organisms.

L337: see comment on cancer

L395: "In all cases..." This is not a result of this study!?

L397: This should sit more at the front of the introduction / discussion.

L427: Not clear why you expect that the corals should show signs of bleaching

Reviewer #2 (Remarks to the Author):

I have now read and considered the manuscript entitled "The impact of Ecological Light Pollution (ELP) on coral reefs in the Gulf of Aqaba/Eilat" for publication in *Communications Biology*. The paper describes an experimental study in which colonies of *Acropora eurystoma* were divided into two groups: control conditions and treatment with light at night, mimicking the exposure to 'ecological light pollution' (ELP). The authors then compared the transcriptome of these two groups and concluded that exposure to ELP lead to differences in gene expression and might induce changes related to cancerous processes in corals.

Overall, I found the article very well-written and consistent. The text is easy to read and a good background is provided in the first section, which helps a lot. I had great fun reading the paper and learned quite a bit (thanks!). Although I'm not a specialist to assess the molecular differences found between the two groups, the results sound pretty convincing given the sheer number of countable differences in genes with known effects. I am aware that this kind of study is eagerly needed, since little is known about genetics for most corals and symbionts, let alone how gene expression changes under different human-induced impacts. Therefore, using a better-known (and important) group like *Acropora* as a model to such pioneering studies sound very sensible to me and it comes at a very critical time for coral conservation, which makes this paper potentially relevant and well-cited. As expected for any study, especially the manipulative ones, in my opinion there are some caveats that I would like the authors to address. They are listed below. However, I think this paper is a very important step towards understanding the actual molecular/physiological consequences of human impacts upon corals.

I have four main considerations.

First, I have concerns in regard to the experimental design itself. I don't think it's fatal to the results, since some differences between groups were very clear, but it does bring some issues to me. Although mesocosms successfully mimic most natural conditions, it's at least conceivable that moving colonies from depths ranging from 4-5 meters in their natural environment to an aquarium at sea level might physiologically affect organisms. And, most importantly, affect individuals differently. There are only five individuals at each treatment (and being from an Ecology background I found this number insufficient to draw definitive conclusions, but I'm aware this is a typical sample size for most molecular studies). Therefore, the final difference in molecular profiles could be due to individual organisms (and that's the problem with small sample sizes) responding differently to those new habitats. From what I understood, sampling only began after the 4-month period of acclimatization, and therefore it is not possible to be sure whether these individuals were already responding differently to their new environments. The assumption that they were under the same conditions and therefore were physiologically similar should have been tested in my opinion, just in case. If they were tested and there are samples from those pre-treatment moments of acclimatization, it would be very useful to compare them and report those results within the paper to reject any possibility of confounding variables affecting these colonies.

Still related to the design, in the future the authors might think of spreading their sampling over the entire period of the experiment, instead of multiple times in one specific day during the period. I think this would give us some better perspective on actual temporal changes. Although the authors correctly state in the Summary that there were "eight time points over a one month period", these samples were all collected on two specific days, four times a day. So, what we have now is a pretty good sampling of two days over a total period of five months (acclimatization + experiment). How beneficial

would it be to have the same eight sampling times more evenly distributed over the whole 5-month period?

Also, I have additional concerns regarding the background ELP that these colonies have been exposed to previously to the study. Although it is not evident from the text that the place they came from was already subjected to ELP (and I would suggest the authors include this information within the manuscript), I presume from Table 1 that there was some ELP exposure in their natural environment. If that's indeed the case, wouldn't it be sensible to include this background ELP into the control group to mimic their natural conditions and expose the treatment group to higher doses of ELP? From Table 1 I get that the control group was not exposed to any light at night at all. If that's true, is it not an indication that the changes in molecular expressions observed over time actually occurred in the control group and not the treatment, since the control group was the one actually subjected to different conditions from their natural environment? Again, had these colonies been sampled during acclimatization would help a lot to clarify who is changing over time.

Finally, although the text is very clear in determining the general outline and goals of the study, it is not always clear what is going to be tested specifically and why. Thus, we see sentences in the Results like "RNA-Seq results were analyzed using the R package GOSEQ (v1.18.0) to detect 256 statistically significantly over-represented functional groups..." (lines 255-256) but it's not clear what were the original questions behind some of these tests. In the above example, for instance, the terms "functional groups" were mentioned for the first time only in those lines at the Results, so it's very hard to see why was this test being performed, let alone their practical importance. Therefore, I suggest the authors could enumerate their specific goals at the end of their Background section, so readers could see what is coming in the results. Moreover, instead of having one subsection named "Statistical analysis" (lines 253-261) which is basically a list of all the tests and R packages necessary for assessing the previous subsections, it would also be more informative to embed what kind of statistical test was used for each objective ("Physiology", "Transcriptome Annotation", "Differential expression" and so on) within their own subsection. The same could be said for the subsection "Pathway analysis".

Beside those bigger considerations I have some minor suggestions:

Line 1: I think the title could be more informative (and attractive) in regard to the most interesting prospects the study brings. For example, I'd suggest the title could point out that light pollution might induce higher expression of cancer-related genes among these corals.

Line 21: Replace "grows" for "grow".

Line 59: Keep acronyms to a minimum. Choose LP or ELP and stick to it throughout. I suggest keeping ELP since LP is actually used just once, here.

Lines 57-61: These two sentences could easily be merged into a more concise and clearer one.

Lines 68-69: Is this a general assertion to all corals or is it more likely to affect those with symbionts? If that's the case, maybe it should be distinguished that you are talking about zooxanthellate corals here.

Line 90: This sentence is confusing. Please rewrite.

Lines 94-97: Although the main goal is established here, it'd be beneficial to state the specific questions/goals of the study.

Line 101: How previously exposed to ELP were these colonies? And how different were these conditions to the RSS conditions? Is it conceivable that the differences observed between two treatments were indeed different reactions to the new environment as a whole? After all, there were only 5 colonies in each treatment. I think this could be tested by sampling and comparing both groups right before the end of the acclimatization period. If any differences were already beginning to show there, maybe there are other explanations to the observed final molecular responses. On the other

hand, if they were not distinguishable in molecular terms after acclimatization, it would be more likely that final differences were due to the effects of the experiment.

Lines 116-118: Did that adjustment include the potential ELP that these corals were exposed to "naturally" in the Gulf of Aqaba? I see in table 1 that PAR exposure at night was similar between open water and the ELP treatment. This should be included in the text to make it clearer to the reader.

Line 223: Species names should be in italics

Line 225: Same as above

Lines 254-255: How was data prepared to t-test? What were the groups of variables being tested at each step? Please be more detailed on your analysis.

Line 256: This is the first (and only) time the terms "functional groups" are mentioned and it's not clear what exactly was done here. Again, it would be easier to understand if all goals were established beforehand so the reader could see what kind of analyses were coming.

Lines 276-280: This first sentence belongs to Discussion.

Line 283: Insert comma before "respectively".

Lines 323-327: This last sentence also belongs to Discussion.

Line 339: I suggest changing "presence of" to "exposure to".

Lines 353-355: Could you elaborate a bit more on this? In the method section it is said that these physiological results would indicate the health status of the corals but by the end of this paragraph it is still not clear if the observed augmented Chlorophyll concentration is good or bad. Intuitively one would guess that if bleaching is bad, more zooxanthellae and/or more Chlorophyll would presumably be good but this paragraph brings alternative and less positive consequences. So it would be good to explain better if, in light of your results, this higher concentrations mean corals more or less healthy.

Line 357: What does "more evolved" here means? Derived?Complex? If so, I strongly suggest changing 'evolved' by any of those other terms.

Lines 377-380: This sentence is a bit confusing since the first part does not mean the opposite of the second, albeit the use of "although". Also, among those studies, is there any registered occurrence of cancer in corals in areas of intense ELP exposure (maybe not measured but at least presumably exposed to ELP)? It would be interesting to find some 'real world' corroboration that reefs near cities (for example) present coral with tumors.

Lines 380-382: I suggest moving this sentence closer to that of lines 375-377 ("Many studies in vertebrates...") since they belong together.

Lines 408-410: Although this is possible, the time scales for which we can infer consequences from the experiment and the time scales involved in reproductive isolation are very different. Therefore, it sounds too speculative. I suggest removing.

Lines 637-638 (Table 1): Given the number of replicates is not very high, I suggest informing both the means and standard deviations.

Signed: Lélis A. Carlos-Júnior

Reviewer #3 (Remarks to the Author):

The manuscript investigates the impact of ecological light pollution, an underestimated anthropogenic pressure, on coral reefs by comparing physiological variables and gene expression between colonies of the hermatypic coral *Acropora eurystroma* maintained under control and light pollution experimental conditions. The authors found that corals under ecological light pollution exhibited significantly more differentially expressed genes, related to cell-cycle regulation, cell proliferation, and other functions related to cancerous processes. This novel study shows that ecological light pollution is an important anthropogenic pressure, with potential deleterious effects that must be recognized and mitigated in order to conserve the health of our coral reefs and other coastal marine ecosystems.

Overall, the manuscript is a very important contribution to the field of cumulative pressures affecting marine ecosystem health, since it demonstrates the negative effect of ecological light pollution in the transcriptome of an ecological engineer, a reef building species. It is well written, although a careful review would be important to correct minor language faults. The introduction is interesting and complete, guiding the reader towards the subject of the study, and based on appropriate references. In the end, a prediction would be important though. The material and methods section is well described in terms of the physiological and transcriptome analysis, yet the sampling design needs more details (see below). The Results are also all right, although I miss more detailing on the number of samples included in the analysis and a general statistic test comparing gene expression among treatments and sampling periods, apart from the heat maps and distance matrices. The Discussion is well conducted, comparing the results with other studies and showing the relevance of the study.

Specific comments:

- I strongly recommend that you include a last sentence in the introduction stating your prediction (Line 97).
- This total number of samples taken is not clear in the text. Perhaps a sampling scheme could help. In the materials and methods section, you state that you collected ten mature colonies of *Acropora eurystoma* and put five of them in each of the two types of aquarium (AMB and ELP). Then you describe that four months later you sampled during four times of the day in two different days (full and new moon). In the manuscript you do not tell how many samples you took, yet in the reporting summary it is stated that all your 10 colonies (5 of each treatment) were sampled at each sampling period? If so, you would have ended with 80 samples, is that right? However, you say that you had 48 samples (Line 48). And when I look at the heatmaps I see 3 samples from each sampling period for ELP (total = 24) and 2-3 samples from each sampling period for AMB (total = 22). What happened to the other samples? It needs to be elucidated in the text.
- Another question, were the samples from a same colony taken close to each other? If not, you could be sampling different genotypes, considering the possibility of chimerism (already shown in other species of *Acropora*). Do you think this could be a confounding effect?
- Add t- values with n and DF, together with p-values (Lines 281-282)
- Was the increase in DE genes detected calculating the mean over the five samples within each treatment? If so, please refer to as : Gene expression patterns revealed a mean increase in (Line 290). In that case standard deviations should be given.
- I really miss a general statistical test (perhaps a glm, but it will depend on your sampling design that is still not completely clear to me, it could also be a split plot) using the number of DE genes as the response variable and Light as a factor (with AMB and ELP as levels) and moon as another factor nested in Light, sampling period nested in moon, and colonies as the error. It would complement the results seen in the heat maps.
- ...to evaluate which experimental variable (Line 293)
- Legend Figure 1 – Normalized (Line 658)
- Legend Figure 3 - purple indicates those genes from one group whose expression pattern resembles... (Line 682)
- Make These light conditions have.... a new sentence. (Line 341)
- will manifest over a longer...(Line 429-430)
- Did you detect any difference in the level of gene expression among samples within the same treatment and in the same period of sampling? This information would be interesting because it could give cues on the existence of genotypic differences in responses to light pollution.

-It is important to recognize that although the number of samples was limited, given xx limitations, the pattern found was strong enough to point to a negative impact of ELP on corals.

-It is also worthy to mention that the study calls attention to a neglected local pressure, and that future work on cumulative pressures carried out in coastal ecosystems should include ecological light pollution.

- Another point to be studied and discussed is whether there is a threshold of light pollution that leads to changes in gene expression in corals and other marine organisms. This knowledge is particularly relevant to guide management decisions on optimal levels of light in coastal cities. Or would the optimal scenario be no light at all? I think it could be important to mention it in the discussion.

-Figures S2 and S3 – I recommend you separate the curves for each treatment, or find a way to represent them in the same curve, to allow that different measures taken at the same time can be visualized with different symbols (maybe symbols with different sizes, one inside the other?). I understand that the curves are superimposed, but the way it is being displayed is quite strange, it seems that measures from different treatments were taken at different times.

Reviewer #1 (Remarks to the Author):

With the rapidly rising urbanisation of coastlines, marine ecosystems are exposed to accordingly increasing levels of artificial light at night (ALAN). As many important physiological processes in marine organisms are regulated and synchronised by solar and lunar cycles, there is a strong concern that Ecological Light Pollution (ELP) may interfere with those mechanisms. This holds true in particular for zooxanthellate reef corals, the habitat-forming species of warm water coral reefs. At present, the effects of ELP on corals are understudied and not well understood. Therefore, the presented manuscript provides important and timely insights in the interference of ELF with gene regulation patterns. Consequently, I am in favour of publication of this manuscript. However, its presentation leaves room for some improvement and I recommend a thorough revision of the paper.

Both, in the introduction and discussion, the logic flow can be improved. At present, the story switches several times between the “bigger/global picture” and specific aspects. Suggest to start with the bigger picture and then zoom in...

There is a strong focus of the manuscript, specifically in the discussion, on differentially regulated genes that are involved in cancer formation. I assume that many differentially regulated genes were identified to be homologous to genes involved in cancer formation in humans /model organisms because of the fact that many signalling pathways are best studied / named in the context of cancer formation. We don't know what the majority of these genes do in corals. While there are some reports on tumors in corals, I would recommend to steer clear from the conclusion that ELP may affect coral via disturbing cell proliferation / cancerogenesis. Instead, a much stronger emphasis should be placed on differentially regulated genes that have been previously studied in corals eg. in the context of spawning, circadian rhythms etc. What is going on with the CRYs?

Because we started sampling after four months of light exposure many of the genes that you are referring to (circadian, spawning) might be asynchronous across the entire month of sampling and because of that we did not find them to be the most differentially expressed. We did mention and gave examples for why the genes we did find to be DE (in regard to cell differentiation, protein synthesis, etc.) become like that. We assume it is all due to the biological clock being out of pace. When we write cancer we are referring to an abnormal cell growth, proliferation and protein synthesis. In a shorter term experiment that we performed we did see many of the genes you are talking about being DE and less genes that are connected to cell cycle like we found here. We think that the longer the exposure to light at night the emphasis is towards the obstruction of the biological clock that leads to cell proliferation. I have rephrased some of the sentences to better explain what we are trying to explain in regards of cancer.

In the introduction and discussion, the use of the experimental ELP intensity should be clearly explained and linked to the environmental data.

Added an explanation at the first part of the method section to better explain why the intensity used to mimic ELP in the aquarium was used “Light intensity in the ELP treatment was adjusted to mimic the same levels of light during the night time that coral reefs in Eilat are exposed to based on Tamir et al., 2017”. We used the measurements done by Raz Tamir in the Gulf to create the same light treatment in our experiment. An additional explanation was added at the first paragraph of the discussion “Corals under ELP treatment were placed under light conditions matching the light pollution experienced by corals in the northern areas in the Gulf of Aquaba/Eilat. An important emphasis was made to create an experimental setup that would resemble the same environmental conditions as the light contaminated areas in the natural reef”.

Figure display could be improved by increasing size/ legibility of axes labels and legend, removing “headers”. In Fig. 4, there could be a better use of space (label / letter size) to improve legibility.

The figure is aimed at benign small and not to take up a full page, the bigger the letter size is the bigger the figure will be.

Figure 5: Suggest to order “Sun rise – Noon – Sun down – Midnight”-

The use of this order is to show the main differences between the sunrise and sundown and noon and midnight as there are the “opposite” light conditions during the day.

The language and spelling of the text should be carefully edited.- **Changed**

The data and the annotation information should be deposited in suitable data base.

All data is deposit at the SRA (accession SRP135621) with all analysis and raw data from the sequencing.

Specific comments:

Line 31: suggest “~25 times” instead of “95%”- **revised**

Line 36: The focus should be taken away from the cancerous processes (See comment above).- **changed to cell proliferation**

Line 45: Include refs.- **added**

L52: re-phrase “dysregulation”- **changed to “disrupting normal circadian rhythms”**

L55: Give examples for the “major implications”- **added “major implications on timing of reproduction, migration, feeding, foraging, sleeping and activity that can all lead lower chances of survival”**

L78: Include reference to blue light regulation of photoprotective pigments in coral host: D'Angelo C, Denzel A, Vogt A, Matz MV, Oswald F, Salih A, Nienhaus GU, Wiedenmann J. 2008Blue light regulation of host pigment in reef-building corals. Mar. Ecol. Progress Ser. 364, 97–106.- **added**

L79: (CRYs) ??? **changed to CRY**

L81: “structuring” does not work in this sentence, also the second half of the sentence is confusing – rephrase.

Changed to: “Since the natural lifecycle of coral reefs is governed by light, there is no doubt that artificial light disturbs their natural processes. Night time lighting has no natural analog- sunlight, moonlight and starlight are the only sources of light corals have adapted to”.

L84-85: Once define, suggest to stick with abbreviations “ALAN” and “ELP”- **changed**

L90-94: Unclear –rework. **Changed to: “The diversity, frequency, and scale of human impacts on coral reefs are increasing to the degree that reefs are threatened worldwide. A major concern is that the rate of environmental change exceeds the evolutionary ability of coral species to adapt. Human caused ELP could alter the natural light regimes of coral reefs by causing persistent disturbance or even chronic stress ”.**

110: Ambient light cycle (AMB). Suggest to call those “control” vs “ELP”. Stay consistent, throughout the ms there are various variations. **Changed.**

Justify the level of ELP used in the experiments clearly in the introduction. **As the levels of ELP mentioned first in the method section I have added an explanation why we choose that specific intensity both in the method and discussion part (see previous mark).**

L131: Give detail of the branch fragments. Where those the top 5cm incl. axial polyp? Further down? **Added: “A small branch from the top of each colony including the axial ,polyp, measuring an average of 5 cm in length, was sampled from each coral using pliers”.**

L139: “electrically” – How? **Added: “Tissue samples were homogenized for 30 s using an electrical tissue homogenizer(Kinematica Polytron™ PT2100 Benchtop Homogenizer) ”**

L154: The RNA extraction is quite detailed and a full protocol is provided in the supplement. This should be shortened, highlighting the parts where the procedure deviated from the manufacturer’s instructions. **Changed.**

L203 onwards: Check the genera / species names are set in italics.- **corrected**

L234: “proteome” ??? **changed to transcriptome**

L288: “de-novo genome” ??? **changed to de-novo transcriptome.**

L.305: “express higher gene resemb.” – rephrase – **changed: “We furthered noticed that in both experimental conditions samples from midnight and sunset had low variance in gene expression and higher variance during the noon sample, i.e. strongest illumination**

throughout the day causes strong variance in gene expression between samples.”

L.316: see comments on cancer

L329: “ecological marine areas” –rephrase.- **changed to “marine areas”**

L334: see comments on cancer

L336: Give examples of the organisms.- **added “ in many organisms from cyanobacteria to humans”**

L337: see comment on cancer

L395: “In all cases...” This is not a result of this study!? **Changed: “Our results suggest that corals possess similar cellular reaction to light pollution as later evolved organisms, the altered light regime caused cell proliferation and organismal abnormalities”.**

L397: This should sit more at the front of the introduction / discussion. **Paragraph moved to the introduction.**

L427: Not clear why you expect that the corals should show signs of bleaching- **Although, it is very low light, the corals are under constant light conditions, we believe that there is probably lower repairing of chlorophyll and the antenna damages. We have now also submitted a paper by this group in which we show that there is more oxidative stress under ELP. All this can lead probably to bleaching.**

Reviewer #2 (Remarks to the Author):

I have now read and considered the manuscript entitled "The impact of Ecological Light Pollution (ELP) on coral reefs in the Gulf of Aqaba/Eilat" for publication in Communications Biology. The paper describes an experimental study in which colonies of *Acropora eurystoma* were divided into two groups: control conditions and treatment with light at night, mimicking the exposure to 'ecological light pollution' (ELP). The authors then compared the transcriptome of these two groups and concluded that exposure to ELP lead to differences in gene expression and might induce changes related to cancerous processes in corals.

Overall, I found the article very well-written and consistent. The text is easy to read and a good background is provided in the first section, which helps a lot. I had great fun reading the paper and learned quite a bit (thanks!). Although I'm not a specialist to assess the molecular differences found between the two groups, the results sound pretty convincing given the sheer number of countable differences in genes with known effects. I am aware that this kind of study is eagerly needed, since little is known about genetics for most corals and symbionts, let alone how gene expression changes under different human-induced impacts. Therefore, using a better-known (and important) group like *Acropora* as a model to such pioneering studies sound very sensible to me and it comes

at a very critical time for coral conservation, which makes this paper potentially relevant and well-cited. As expected for any study, especially the manipulative ones, in my opinion there are some caveats that I would like the authors to address. They are listed below. However, I think this paper is a very important step towards understanding the actual molecular/physiological consequences of human impacts upon corals.

I have four main considerations.

First, I have concerns in regard to the experimental design itself. I don't think it's fatal to the results, since some differences between groups were very clear, but it does bring some issues to me. Although mesocosms successfully mimic most natural conditions, it's at least conceivable that moving colonies from depths ranging from 4-5 meters in their natural environment to an aquarium at sea level might physiologically affect organisms. And, most importantly, affect individuals differently. There are only five individuals at each treatment (and being from an Ecology background I found this number insufficient to draw definitive conclusions, but I'm aware this is a typical sample size for most molecular studies). Therefore, the final difference in molecular profiles could be due to individual organisms (and that's the problem with small sample sizes) responding differently to those new habitats. From what I understood, sampling only began after the 4-month period of acclimatization, and therefore it is not possible to be sure whether these individuals were already responding differently to their new environments. The assumption that they were under the same conditions and therefore were physiologically similar should have been tested in my opinion, just in case. If they were tested and there are samples from those pre-treatment moments of acclimatization, it would be very useful to compare them and report those results within the paper to reject any possibility of confounding variables affecting these colonies.

All coral colonies were collected from the same location and the same depth. Their original location (I.U.I coral nursery) had no light contamination during the night so at the start of the experiment they were all considered AMB corals. The aquariums were prepared in advance with filters covering to adjust the light the experience to be the same as the natural reef they were collected from. From our physiology assay results and RNA sequencing we found that corals from the same treatment reacted the same and clustered together as opposed to the other group. As in any experiment there were individual variation resulting from the different colonies but they were minor and did not affect our results. When summing the expression of each group it was completely different than the other group. Sampling started after four months in order to see the cellular reaction of corals to light pollution and not in a short term experiment. We did not sample at the start of the experiment, we used the AMB colonies as reference for the effect of light on corals. From other studies we have done we saw the same trend in physiology parameters that we saw in our AMB corals and therefore we think that the AMB corals results represent the natural and normal results of corals that are not under light pollution and could be used as a good comparison in our study. We used five individuals in each treatment as that is the number of colonies we could use in the experiment from the Israeli nature and parks authority. We also know from previous preliminary experiment that the response to light pollution is chronic and not aquatic.

Which means it takes time to see differences like we point in this paper. We hope this point it clearer now.

Still related to the desing, in the future the authors might think of spreading their sampling over the entire period of the experiment, instead of multiple times in one specific day during the period. I think this would give us some better perspective on actual temporal changes. Although the authors correctly state in the Summary that there were "eight time points over a one month period", these samples were all collected on two specific days, four times a day. So, what we have now is a pretty good sampling of two days over a total period of five months (acclimatization + experiment). How beneficial would it be to have the same eight sampling times more evenly distributed over the whole 5-month period?

Our sampling regime was aimed at finding genes that are differentially expressed across a daily cycle and a moon phase cycle. To address that issue, we sample four times a day to find variation during the day and sampled on new moon and full moon to see the differences between those night specifically. The total time of the experiment was to allow the corals to first acclimate to the conditions and secondly to understand how the light affects the corals after a considerable period. The goal of the experiment was to find specific cellular pathways that change between the treatments daily and between moon phases and not across the year. As for longer time, we are now working on a new manuscript where we have sampled annual cycle, diel cycle and monthly cycle including more than 130 RNA-seq libraries including physiology and stable isotopes analysis. This future manuscript will deal deeper into biological cycles and will serve as a complementary to this present work.

Also, I have additional concerns regarding the background ELP that these colonies have been exposed to previously to the study. Although it is not evident from the text that the place they came from was already subjected to ELP (and I would suggest the authors include this information within the manuscript), I presume from Table 1 that there was some ELP exposure in their natural environment. If that's indeed the case, wouldn't it be sensible to include this background ELP into the control group to mimic their natural conditions and expose the treatment group to higher doses of ELP? From Table 1 I get that the control group was not exposed to any light at night at all. If that's true, is it not an indication that the changes in molecular expressions observed over time actually occurred in the control group and not the treatment, since the control group was the one actually subjected to different conditions from their natural environment? Again, had these colonies been sampled during acclimatization would help a lot to clarify who is changing over time.

All coral colonies were collected from the same location and the same depth. Their original location (I.U.I coral nursery) had no light contamination during the night so at the start of the experiment they were all considered AMB corals. The corals that were placed under ELP treatment were the experimental group and compared to the AMB corals that did not have different light exposer from their original habitat. We did not sample during acclimatization as sequencing costs are high we wanted to see the differences between the AMB and ELP corals and used the AMB corals as a baseline.

Finally, although the text is very clear in determining the general outline and goals of the study, it is not always clear what is going to be tested specifically and why. Thus, we see sentences in the Results like "RNA-Seq results were analyzed using the R package GOSEQ (v1.18.0) to detect 256 statistically significantly over-represented functional groups..." (lines 255-256) but it's not clear what were the original questions behind some of these tests. In the above example, for instance, the terms "functional groups" **(changed to: "statistically significantly over-represented genes that cluster into functional groups)** were mentioned for the first time only in those lines at the Results, so it's very hard to see why was this test being performed, let alone their practical importance **(added: "when comparing between samples from both experimental groups).** Therefore, I suggest the authors could enumerate their specific goals at the end of their Background section, so readers could see what is coming in the results. Moreover, instead of having one subsection named "Statistical analysis" (lines 253-261) which is basically a list of all the tests and R packages necessary for assessing the previous subsections, it would also be more informative to embed what kind of statistical test was used for each objective ("Physiology", "Transcriptome Annotation", "Differential expression" and so on) within their own subsection. The same could be said for the subsection "Pathway analysis". **Statistical analysis was embedded in each section in the method.**

Beside those bigger considerations I have some minor suggestions:

Line 1: I think the title could be more informative (and attractive) in regard to the most interesting prospects the study brings. For example, I'd suggest the title could point out that light pollution might induce higher expression of cancer-related genes among these corals. **We think you are right, but as reviewer 1 pointed, he does not want us to use the phrase cancer as much. So we think our title is a compromise between the two of you.**
in Line 21: Replace "grows" for "grow". - **changed**

Line 59: Keep acronyms to a minimum. Choose LP or ELP and stick to it throughout. I suggest keeping ELP since LP is actually used just once, here. **Changed all phrases to ELP and left LP only once in this line.**

Lines 57-61: These two sentences could easily be merged into a more concise and clearer one.- **changed to one sentence: "The introduction of artificial lighting, particularly electric lighting, has disrupted natural cycles of light and darkness and is now referred to as "light pollution" (LP) or "Ecological light pollution" (ELP), a term that has been coined to describe all types of artificial light that alter the natural patterns of light and dark in ecosystems"**

Lines 68-69: Is this a general assertion to all corals or is it more likely to affect those with symbionts? If that's the case, maybe it should be distinguished that you are talking about zooxanthellate corals here. **changed to Zooxanthellate corals.**

Line 90: This sentence is confusing. Please rewrite. **Changed: "The diversity, frequency, and scale of human impacts on coral reefs are increasing to the degree that reefs are threatened worldwide. A major concern is that the rate of environmental change exceeds the evolutionary ability of coral species to adapt"**.

Lines 94-97: Although the main goal is established here, it'd be beneficial to state the specific questions/goals of the study. **Added: ". We aimed to reveal specific cellular pathways and genes that cluster into functional groups which change between the treatments in response to the daily and moon phase cycles. Finding the variations in**

gene expression caused by the different light regime could help better understand the effect of light at night on corals life cycle.”

Line 101: How previously exposed to ELP were these colonies? And how different were these conditions to the RSS conditions? Is it conceivable that the differences observed between two treatments were indeed different reactions to the new environment as a whole? After all, there were only 5 colonies in each treatment. I think this could be tested by sampling and comparing both groups right before the end of the acclimatization period. If any differences were already beginning to show there, maybe there are other explanations to the observed final molecular responses. On the other hand, if they were not distinguishable in molecular terms after acclimatization, it would be more likely that final differences were due to the effects of the experiment. – **added an explanation in the first part of the methods explaining where the corals were collected from and mentioning they had no exposure to light at night prior to the experiment.**

Lines 116-118: Did that adjustment include the potential ELP that these corals were exposed to "naturally" in the Gulf of Aqaba? I see in table 1 that PAR exposure at night was similar between open water and the ELP treatment. This should be included in the text to make it clearer to the reader. **I added a sentence explaining that in certain parts of the Gulf there is light contamination at night (as the levels in table 1). There are little parts in the gulf with no light exposure, like I.U.I where the corals were retrieved from.**

Line 223: Species names should be in italics- **changed**

Line 225: Same as above- **changed**

Lines 254-255: How was data prepared to t-test? What were the groups of variables being tested at each step? Please be more detailed on your analysis. **Added: “Each result is an average of five fragments per treatment. Physiology assay results of both treatments were compared to each other in each parameter using the R package “unpaired two-samples t-test” to represent statistical relevance. “**

Line 256: This is the first (and only) time the terms "functional groups" are mentioned and it's not clear what exactly was done here. Again, it would be easier to understand if all goals were established beforehand so the reader could see what kind of analyses were coming. **Added a paragraph at the end of the introduction explaining our goal (see above) and added in the text:” detect statistically significantly over-represented genes that cluster into functional groups”.** If it is still not clear I would be happy to change the phrase.

Lines 276-280: **This first sentence belongs to Discussion. This sentence was placed here in order to give an introduction to the results.**

Line 283: Insert comma before "respectively". **added**

Lines 323-327: This last sentence also belongs to Discussion. **moved to discussion part.**

Line 339: I suggest changing "presence of" to "exposure to". **Changed.**

Lines 353-355: Could you elaborate a bit more on this? In the method section it is said that these physiological results would indicate the health status of the corals but by the end of this paragraph it is still not clear if the observed augmented Chlorophyll concentration is good or bad. Intuitively one would guess that if bleaching is bad, more zooxanthellae and/or more Chlorophyll would presumably be good but this paragraph brings alternative and less positive consequences. So it would be good to explain better if, in light of your results, this higher concentrations mean corals more or less healthy. **As there is no prior research on symbiotic corals under light pollution (there are**

studies on constant light but with no changes in light phases like we have done here) we cannot determine the consequence of the elevated chlorophyll we have observed. As mentioned in the manuscript we assume these results are due to the obstruction of the biological clock of the algae that causes the production of more chlorophyll per algae cell. Any changes to the normal cell cycle could be harmful over time but we cannot state if it's healthy or not. We simply wanted to point out that photo-physiology can alter due to light pollution, in this case it is the chlorophyll changes.

Line 357: What does "more evolved" here mean? Derived? Complex? If so, I strongly suggest changing 'evolved' by any of those other terms. **Changed to more complex.**

Lines 377-380: This sentence is a bit confusing since the first part does not mean the opposite of the second, albeit the use of "although". Also, among those studies, is there any registered occurrence of cancer in corals in areas of intense ELP exposure (maybe not measured but at least presumably exposed to ELP)? It would be interesting to find some 'real world' corroboration that reefs near cities (for example) present coral with tumors. **Light pollution research is fairly new and not many have been conducted on "non-model" organisms. Most studies regarding coral tumor formation was not related to light exposure but rather relating their immune system.**

Lines 380-382: I suggest moving this sentence closer to that of lines 375-377 ("Many studies in vertebrates...") since they belong together. **Moved**

Lines 408-410: Although this is possible, the time scales for which we can infer consequences from the experiment and the time scales involved in reproductive isolation are very different. Therefore, it sounds too speculative. I suggest removing. **There are different papers relating to asynchronous planula release because of a different light regime. Here we state that there should be no different consequence when exposed to light pollution as it also changes the time perception of the corals and masks the moon light. Sadly, there is no change in light exposure across the world and it is only becoming a bigger problem, therefore, we think that over time it could drive to reproductive isolation and prevent gene flow.**

Lines 637-638 (Table 1): Given the number of replicates is not very high, I suggest informing both the means and standard deviations. – **changed**

Signed: Lélis A. Carlos-Júnior

Reviewer #3 (Remarks to the Author):

The manuscript investigates the impact of ecological light pollution, an underestimated anthropogenic pressure, on coral reefs by comparing physiological variables and gene expression between colonies of the hermatypic coral *Acropora eurystoma* maintained under control and light pollution experimental conditions. The authors found that corals under ecological light pollution exhibited significantly more differentially expressed genes, related to cell-cycle regulation, cell proliferation, and other functions related to cancerous processes. This novel study shows that ecological light pollution is an

important anthropogenic pressure, with potential deleterious effects that must be recognized and mitigated in order to conserve the health of our coral reefs and other coastal marine ecosystems.

Overall, the manuscript is a very important contribution to the field of cumulative pressures affecting marine ecosystem health, since it demonstrates the negative effect of ecological light pollution in the transcriptome of an ecological engineer, a reef building species. It is well written, although a careful review would be important to correct minor language faults. The introduction is interesting and complete, guiding the reader towards the subject of the study, and based on appropriate references. In the end, a prediction would be important though. The material and methods section is well described in terms of the physiological and transcriptome analysis, yet the sampling design needs more details (see below). The Results are also all right, although I miss more detailing on the number of samples included in the analysis and a general statistic test comparing gene expression among treatments and sampling periods, apart from the heat maps and distance matrices. The Discussion is well conducted, comparing the results with other studies and showing the relevance of the study.

Specific comments:

-I strongly recommend that you include a last sentence in the introduction stating your prediction (Line 97). **Added: “. We aimed to reveal specific cellular pathways and genes that cluster into functional groups which change between the treatments in response to the daily and moon phase cycles. Finding the variations in gene expression caused by the different light regime could help better understand the effect of light at night on corals life cycle.”**

- This total number of samples taken is not clear in the text. Perhaps a sampling scheme could help. In the materials and methods section, you state that you collected ten mature colonies of *Acropora eurystoma* and put five of them in each of the two types of aquarium (AMB and ELP). Then you describe that four months later you sampled during four times of the day in two different days (full and new moon). In the manuscript you do not tell how many samples you took, yet in the reporting summary it is stated that all your 10 colonies (5 of each treatment) were sampled at each sampling period? If so, you would have ended with 80 samples, is that right? However, you say that you had 48 samples (Line 48). And when I look at the heatmaps I see 3 samples from each sampling period for ELP (total = 24) and 2-3 samples from each sampling period for AMB (total = 22). What happened to the other samples? **It needs to be elucidated in the text. I have added a sentence in the method section describing that only three replicates were sent for sequencing. Sequencing is highly expensive and we choose three samples from each treatment to use for RNA-Seq. in the Heatmap there are 2 samples from the AMB treatment (sunset and midnight of the full moon) that we excluded from the analysis due to technical problems with the sequencing process.**

-Another question, were the samples from a same colony taken close to each other? If not, you could be sampling different genotypes, considering the possibility of chimerism (already shown in other species of *Acropora*). Do you think this could be a confounding

effect? **The sampling area on each coral as taken randomly. After RNA-Seq we could see each sample clustering together with the other samples from the same colony so the “gene flow” was similar based on the colony itself.**

-Add t- values with n and DF, together with p-values (Lines 281-282) – **added t-values with DF values.**

-Was the increase in DE genes detected calculating the mean over the five samples within each treatment? If so, please refer to as : Gene expression patterns revealed a mean increase in (Line 290). In that case standard deviations should be given. **The p value mentioned here is not the mean of all genes from a treatment but the threshold of what is considered a differentially expressed gene. Meaning only if its p-value when compared to the other genes in the transcriptome and to its length is lower than 0.05 and its fold change when compared to the AMB gene group is higher than 1.5.**

-I really miss a general statistical test (perhaps a glm, but it will depend on your sampling design that is still not completely clear to me, it could also be a split plot) using the number of DE genes as the response variable and Light as a factor (with AMB and ELP as levels) and moon as another factor nested in Light, sampling period nested in moon, and colonies as the error. It would complement the results seen in the heat maps.**We have listed in the text regarding our statistical analysis of the gene expression. This statistical analysis was previously published in many papers from our group and others and it is based on the sampling design we have used in this work.**

-...to evaluate which experimental variable (Line 293) **changed**

-Legend Figure 1 – Normalized (Line 658) – **changed**

-Legend Figure 3 - purple indicates those genes from one group whose expression pattern resembles... (Line 682) - **changed**

-Make These light conditions have.... a new sentence. (Line 341) –**changed**

-will manifest over a longer...(Line 429-430) – **changed**

-Did you detect any difference in the level of gene expression among samples within the same treatment and in the same period of sampling? This information would be interesting because it could give cues on the existence of genotypic differences in responses to light pollution. **As mentioned before After RNA-Seq we could see each sample clustering together with the other samples from the same colony so the “gene flow” was similar based on the colony itself. There are differences in gene expression resulting from the different colonies but they all showed the same pattern and when normalized to the whole transcriptome the differences were minor.**

-It is important to recognize that although the number of samples was limited, given xx limitations, the pattern found was strong enough to point to a negative impact of ELP on corals.

-It is also worthy to mention that the study calls attention to a neglected local pressure, and that future work on cumulative pressures carried out in coastal ecosystems should include ecological light pollution. – **it is now mentioned in the last sentence of the discussion.**

- Another point to be studied and discussed is whether there is a threshold of light pollution that leads to changes in gene expression in corals and other marine organisms. This knowledge is particularly relevant to guide management decisions on optimal levels

of light in coastal cities. Or would the optimal scenario be no light at all? I think it could be important to mention it in the discussion. **Added: “We will continue with studies that will help determine the specific threshold of light intensity and wavelength that effect corals in order to guide management decisions on optimal levels of light in coastal cities.”**

-Figures S2 and S3 – I recommend you separate the curves for each treatment, or find a way to represent them in the same curve, to allow that different measures taken at the same time can be visualized with different symbols (maybe symbols with different sizes, one inside the other?). I understand that the curves are superimposed, but the way it is being displayed is quite strange, it seems that measures from different treatments were taken at different times. **Measurements were taken at the same time and had the exact same values, when separating the lines, we get one covering the other, that is why it is presented like this, I hope it is understandable.**

REVIEWERS' COMMENTS:

Reviewer #2 (Remarks to the Author):

After the authors' considerations I am satisfied with most of the issues I raised during the first round of reviews. Therefore, I agree the paper is mostly suited for publication, provided they address some additional comments, as listed below.

1) Although the authors explained within the new version of their text how there was no ELP at the reef the corals were originally extracted, I still suggest that it would be beneficial in future studies to establish (rather than trust by default) that colonies under both treatments were responding equally to the mesocosms environment after the acclimatisation period. My concern sits mainly due to the low number of colonies per treatment, which I understand is not always ideal due to logistics, costs, authorisations, etc. For the current paper, however, the authors gave me some reasonable evidence that corals in both groups should be physiologically similar after the 4-month period in their rebuttal letter: "We did not sample at the start of the experiment, we used the AMB colonies as reference for the effect of light on corals. From other studies we have done we saw the same trend in physiology parameters that we saw in our AMB corals and therefore we think that the AMB corals results represent the natural and normal results of corals that are not under light pollution and could be used as a good comparison in our study.". I strongly suggest this should be included within the manuscript (somewhere around lines 212-213 in the new version with tracked changes), citing these other studies from the group if possible, to avoid other readers having the same doubts I had when I first read the ms.

As I stated in my first review, I do not think these considerations on sampling during acclimatisation were fatal to their current results, since the large number of differences found at the end hints that indeed some very different stuff was going on between groups. Nevertheless, this additional sample should be useful in the future to spot more subtle variations among treatment and control groups.

2) Since I do not come from a molecular ecology background I had suggested some changes to the title concerning the cancer-relatedness of the observed changes in transcriptomes among those corals. However, as alerted by reviewer #1, I agree that more caution should be taken in the general phrasing of this aspect throughout the paper. Below, I give several suggestions on where those cautionary comments could be inserted.

Minor considerations:

- Line 179 (and on other portions): nighttime (or night-time) instead of night time.
- Line 199: What does AMB stand for? I could not find its actual meaning within the text.
- Line 274: I suggest including total number of samples (N=?) at the end of sentence.
- Line 351: Species name should be italicised.
- Lines 473-475: Include the model organisms in which such pathways were studied (e.g. they are related to cancer in which kind of animal?).
- lines 478-485: same as above.
- Line 492-493: "important emphasis" sounds very redundant.
- Line 504: "exposure" instead of "exposer"?
- Lines 570-572: Again, be more specific on which kind of organisms were studied here.
- Lines 579-580: Same as above

Lélis A Carlos Júnior

Reviewer #3 (Remarks to the Author):

I have now carefully read the authors reply to the referees reviews as well as the revised version of the manuscript "The impact of Ecological Light Pollution (ELP) on coral reefs in the Gulf of Aqaba/Eilat". As I have previously stated, the manuscript is a very important contribution to the field of cumulative pressures affecting marine ecosystem health, since it demonstrates the negative effect of ecological light pollution in the transcriptome of an ecological engineer, a reef building species.

Overall, the authors paid attention to the points raised by the referees, justifying their choices properly in some cases or making the requested changes in the manuscript in other cases. They also recognised the limitations of the study in terms of number of replicates and experiment duration. Every study has its own limitation, however it is important to recognise it and to explain what was done to overcome the limitation and/or how it influences (or not) the interpretation of the results found.

I am satisfied with the answers and eventual changes related to my questions and recommendations to the first version of the manuscript.

I have only two minor specific remarks:

Line 433 - Replace consist by perform

Lines 434-437 - Please rewrite, the sentence is long and confusing.

REVIEWERS' COMMENTS:

Reviewer #2 (Remarks to the Author):

After the authors' considerations I am satisfied with most of the issues I raised during the first round of reviews. Therefore, I agree the paper is mostly suited for publication, provided they address some additional comments, as listed below.

1) Although the authors explained within the new version of their text how there was no ELP at the reef the corals were originally extracted, I still suggest that it would be beneficial in future studies to establish (rather than trust by default) that colonies under both treatments were responding equally to the mesocosms environment after the acclimatisation period. My concern sits mainly due to the low number of colonies per treatment, which I understand is not always ideal due to logistics, costs, authorisations, etc. For the current paper, however, the authors gave me some reasonable evidence that corals in both groups should be physiologically similar after the 4-month period in their rebuttal letter: "We did not sample at the start of the experiment, we used the AMB colonies as reference for the effect of light on corals. From other studies we have done we saw the same trend in physiology parameters that we saw in our AMB corals and therefore we think that the AMB corals results represent the natural and normal results of corals that are not under light pollution and could be used as a good comparison in our study.". I strongly suggest this should be included within the manuscript (somewhere around lines 212-213 in the new version with tracked changes), citing these other studies from the group if possible, to avoid other readers having the same doubts I had when I first read the ms.

As I stated in my first review, I do not think these considerations on sampling during acclimatisation were fatal to their current results, since the large number of differences found at the end hints that indeed some very different stuff was going on between groups. Nevertheless, this additional sample should be useful in the future to spot more subtle variations among treatment and control groups.

2) Since I do not come from a molecular ecology background I had suggested some changes to the title concerning the cancer-relatedness of the observed changes in transcriptomes among those corals. However, as alerted by reviewer #1, I agree that more caution should be taken in the general phrasing of this aspect throughout the paper. Below, I give several suggestions on where those cautionary comments could be inserted.

Minor considerations:

- Line 179 (and on other portions): nighttime (or night-time) instead of night time. **Changed to nighttime**
- Line 199: What does AMB stand for? I could not find its actual meaning within the text. **Added Ambient- AMB corals in the text**
- Line 274: I suggest including total number of samples (N=?) at the end of sentence. **added n=5 corals per treatment.**
- Line 351: Species name should be italicised. **changed**
- Lines 473-475: Include the model organisms in which such pathways were studied (e.g. they are related to cancer in which kind of animal?). **sentence starts with "Many studies in**

vertebrates”

- lines 478-485: same as above.
- Line 492-493: "important emphasis" sounds very redundant. **Changed to emphasis.**
- Line 504: "exposure" instead of "exposer"? **changed**
- Lines 570-572: Again, be more specific on which kind of organisms were studied here.
- Lines 579-580: Same as above

Lélis A Carlos Júnior

Reviewer #3 (Remarks to the Author):

I have now carefully read the authors reply to the referees reviews as well as the revised version of the manuscript "The impact of Ecological Light Pollution (ELP) on coral reefs in the Gulf of Aqaba/Eilat". As I have previously stated, the manuscript is a very important contribution to the field of cumulative pressures affecting marine ecosystem health, since it demonstrates the negative effect of ecological light pollution in the transcriptome of an ecological engineer, a reef building species.

Overall, the authors paid attention to the points raised by the referees, justifying their choices properly in some cases or making the requested changes in the manuscript in other cases. They also recognised the limitations of the study in terms of number of replicates and experiment duration. Every study has its own limitation, however it is important to recognise it and to explain what was done to overcome the limitation and/or how it influences (or not) the interpretation of the results found.

I am satisfied with the answers and eventual changes related to my questions and recommendations to the first version of the manuscript.

I have only two minor specific remarks:

Line 433 - Replace consist by perform. **changed**

Lines 434-437 - Please rewrite, the sentence is long and confusing. **re-written**